# Pleiotropic and Potentially Beneficial Effects of Reactive Oxygen Species on the Intracellular Signaling Pathways in Endothelial Cells

**DOI:** 10.3390/antiox10060904

**Published:** 2021-06-03

**Authors:** Nadezhda Barvitenko, Elisaveta Skverchinskaya, Alfons Lawen, Elena Matteucci, Carlota Saldanha, Giuseppe Uras, Alessia Manca, Muhammad Aslam, Antonella Pantaleo

**Affiliations:** 1Independent Researcher, 191014 Saint-Petersburg, Russia; 2Sechenov Institute of Evolutionary Physiology and Biochemistry, 194223 Saint-Petersburg, Russia; lisarafail@mail.ru; 3Department of Biochemistry and Molecular Biology, School of Biomedical Sciences, Monash University, Melbourne, VIC 3800, Australia; alfons.lawen@monash.edu; 4Department of Clinical and Experimental Medicine, University of Pisa, Via Roma 67, 56126 Pisa, Italy; elena.matteucci@med.unipi.it; 5Institute of Biochemistry, Institute of Molecular Medicine, Faculty of Medicine University of Lisbon, 1649-028 Lisboa, Portugal; carlotasaldanha@fm.ul.pt; 6Department of Clinical and Movement Neurosciences, Institute of Neurology, University College London, London NW3 2PF, UK; g.uras@ucl.ac.uk; 7Department of Biomedical Science, University of Sassari, Viale San Pietro 43/B, 07100 Sassari, Italy; alessia_manca@hotmail.it; 8Experimental Cardiology, Justus Liebig University, 35392 Giessen, Germany; muhammad.aslam@physiomed.jlug.de

**Keywords:** reactive oxygen species, endothelial cell, insulin resistance, endothelial paracellular permeability, endothelial dysfunction

## Abstract

Endothelial cells (ECs) are exposed to molecular dioxygen and its derivative reactive oxygen species (ROS). ROS are now well established as important signaling messengers. Excessive production of ROS, however, results in oxidative stress, a significant contributor to the development of numerous diseases. Here, we analyze the experimental data and theoretical concepts concerning positive pro-survival effects of ROS on signaling pathways in endothelial cells (ECs). Our analysis of the available experimental data suggests possible positive roles of ROS in induction of pro-survival pathways, downstream of the G_i_-protein-coupled receptors, which mimics insulin signaling and prevention or improvement of the endothelial dysfunction. It is, however, doubtful, whether ROS can contribute to the stabilization of the endothelial barrier.

## 1. Introduction

Cells of vasculature, red blood cells (RBCs), endothelial cells (ECs), and vascular smooth muscle cells (VSMCs), work in concert to match the oxygen supply with tissue oxygen demand [1]. Some oxygen molecules encounter free electrons (e^−^) and free protons (H^+^), resulting in the formation of reactive oxygen species (ROS). ROS include superoxide anion (O_2_^●^^−^), hydrogen peroxide (H_2_O_2_), and hydroxyl radical (^●^OH). The reaction of superoxide anion with nitric oxide (NO) produces peroxynitrite (ONOO^−^). There are multiple sources of ROS in ECs, including nicotinamide adenine dinucleotide phosphate (NADPH) oxidases, xanthine oxidases, nitric oxide synthase (NOS), cyclooxygenase (COX), cytochrome P450 monooxygenases, and mitochondria [2,3]. There are seven members of the NADPH oxidase (Nox) family—Nox1, Nox2 [also known as gp91phox (phox stands for phagocyte oxidase)], Nox3, Nox4, Nox5, Duox1 (dual oxidase), and Duox2 [4]. Nox1, Nox2, Nox4, and Nox5 isoforms are expressed in cells of the cardiovascular system [3,5,6,7].

ROS are important signaling molecules that can influence various signaling proteins and contribute to cell survival [8,9]. Protein tyrosine phosphatases are reversibly inhibited by ROS [10,11,12]. Many protein kinases can be regulated by ROS, including Src family tyrosine kinases, receptor tyrosine kinases, c-Abl tyrosine kinase, Akt, cAMP-dependent protein kinase (PKA), mitogen-activated protein kinases (MAPKs), Ca^2+^/calmodulin-dependent protein kinase II (CaMKII), cGMP-dependent protein kinase Iα (PKGIα), ataxia-telangiectasia mutated (ATM) protein kinase, and apoptosis signal-regulated kinase 1 (ASK1) [13,14].

Excessive ROS production is one of the major causes of hypertension [15], atherosclerosis [16], and other cardiovascular diseases [17], i.e., pathological states that depend on endothelial dysfunction. Endothelial dysfunction itself can result from oxidative stress [18]. However, accumulating data suggest that ROS at the physiological level perform pro-survival roles in ECs [5,6,7,19,20,21,22]. This diversity and multiplicity of signaling proteins that can be directly regulated by ROS, prompted us to further analyze if, and how, ROS may prevent or improve some pathological conditions of the endothelium, such as insulin resistance, disruption of the endothelial barrier, and endothelial dysfunction. A hypothesis on the integration of the signaling pathways by microtubules [23] may help in understanding the control of signaling networks by ROS.

In this review, we analyze the available experimental data on activation of pro-survival signaling pathways in ECs by ROS. Since the hydroxyl radical produced from H_2_O_2_ can directly activate inhibitory α subunits (Gα_i/o_) of heterotrimeric G proteins [24,25], we first consider if ROS can mimic downstream signaling of G_i_-protein-coupled receptors (G_i_-PCRs) (Section 2) and insulin signaling (Section 3). Next, we arrange the roles of ROS in accordance with their hypothetically protective roles in pathological states of endothelium, such as endothelial barrier disruption (Section 4), endothelial dysfunction (Section 5), and angiogenesis (Section 6).

## 2. ROS, Gα_i/o_ Subunits of the Heterotrimeric G Proteins and EC Survival

There are four families of the α subunits of the heterotrimeric G proteins—Gα*_s_*, Gα*_i/o_*, Gα*_q/11_*, and Gα*_12/13_* [26]. In neonatal rat, the Gα_i/o_ subunits of cardiomyocytes were shown to be directly activated by hydroxyl radicals produced from H_2_O_2_, in the presence of Fe^2+^ [24,25]. Among the seven cysteine residues present in Gα_i2_ (Cys66, Cys112, Cys140, Cys255, Cys287, Cys326, and Cys352), Cys287 and Cys326 are responsible for Gα_i2_ activation by hydroxyl radicals [25].

Apoptosis induced by high glucose in human pancreatic islet microvascular ECs can be inhibited by activation of the phosphatidylinositol 3-kinase (PI3K)–Akt, extracellular signal-regulated kinase (ERK) 1/2, and adenylyl cyclase (AC)–cAMP–PKA pathways [27]. Here, we regard the potential role of G_i_ proteins in these three pro-survival pathways in ECs: AC–cAMP–PKA–cAMP response element-binding protein (CREB), Ras–Raf–mitogen-activated protein kinase/extracellular signal-regulated kinase 1/2 (MEK1/2)–ERK1/2, and PI3K–Akt (Figure 1).

As discussed below, there is experimental evidence for activation of these pro-survival pathways by G_i_-protein-coupled receptors (G_i_-PCRs) via liberation and activation of the Gβγ dimers (Figure 1). Activation of Gα_i/o_ by ROS would also activate Gβγ [24,25]. This makes us suggest that ROS via activation of G_i_ proteins can promote a G_i_-PCR-independent EC survival. Data on triggering of EC survival by some G_i_-PCRs are presented in Table 1. However, activation of Gi-PCRs above a specific threshold may also induce apoptosis (Table 1).

### 2.1. G_i/o_ Proteins and AC–cAMP–PKA Pathway

AC activation via production of cAMP and activation of PKA leads to phosphorylation and nuclear translocation of CREB [36]. CREB can mediate the pro-survival effect of the AC–cAMP–PKA pathway. For example, in mouse, the immortalized cerebral endothelial (b.End3) cells CREB is responsible for vascular endothelial growth factor A (VEGF-A)/VEGF receptor-2 (VEGFR-2)-mediated cell survival [37]. In human umbilical vein, endothelial cells (HUVECs) lipopolysaccharide (LPS)-induced apoptosis was inhibited by cilostazol, a selective phosphodiesterase 3 inhibitor, via increase in cAMP, activation of MEK1/2–ERK1/2 and p38, and activation of CREB [38].

There can be signaling from the G_i_-coupled receptors to CREB activation, although the Gα_i_ subunits inhibit AC. For instance, the relaxin family peptide receptor 1 (RXFP1) is coupled with Gα_i3_ and can activate the Gβγ–PI3K–PKCζ (protein kinase Czeta)–AC pathway [39] (Figure 1). Survival of human pancreatic islet microvascular ECs was enhanced by gastrointestinal ghrelin gene products acting via their receptor GHS-R1a (growth hormone secretagogue receptor 1a), which induced activation of the AC–cAMP–PKA pathway [27] (Table 1). GHS-R1a is mainly coupled to Gα_q/11_ but also to G_i_ [33].

### 2.2. G_i/o_ Proteins and Ras–Raf–MEK1/2–ERK1/2 Pathway

A cell’s choice between survival and apoptosis can be determined by the balance between activities of members of the family of mitogen-activated protein kinases (MAPKs)—ERK, JNK (c-Jun NH_2_-terminal protein kinase), and p38 MAPK [40]. It appears that apoptosis can be promoted by activation of JNK and p38, accompanied by inhibition of ERK [40]. Hyperglycemia-induced apoptosis in ECs can be inhibited by activation of ERK1/2 [27]. Like other cells, in ECs, ERK1/2 is an element of the Ras–Raf-1–MEK1/2–ERK1/2 pathway [41]. Direct activation of G_i_ proteins by ROS induces ERK1/2 activation [24,25]. How can activation of G_i_ proteins lead to activation of the Ras–Raf–MEK1/2–ERK1/2 pathway? Sphingosine-1-phosphate (S1P), a platelet-derived phospholipid, binds to its receptors, presented by five isotypes—S1P_1_ [also known as Edg1 (endothelial differentiation gene 1)], S1P_2_ (Edg5), S1P_3_ (Edg3), S1P_4_ (Edg6), and S1P_5_ (Edg8) receptors [34]. Among these receptors, S1P_1_ is exclusively coupled to G_i/o_ proteins [34]. In HUVECs and HEK293 cells, the G_i_ proteins downstream of Edg-1 (S1P_1_ receptor) were shown to activate ERK-2 and promote survival [42]. Activation of ERK1/2 by the G_i_ proteins [35,42] may result from activation of the small GTPase Ras, by Gβγ dimers dissociated from the Gα_i_ subunit [43,44] (Figure 1). Thus, the Gβγ dimers, which can be activated via ROS-induced activation of Gαi [24,25] are likely to activate the Ras–Raf-1–MEK1/2–ERK1/2 cascade in a receptor-independent manner, and to promote EC survival (Figure 1).

### 2.3. G_i/o_ Proteins and the PI3K–Akt Pathway

The activation of the PI3K–Akt is a well-established pro-survival pathway [45,46,47]. In human pulmonary artery ECs (HPAECs), Gβγ dimers when dissociated from the G_i_-coupled S1P_1_ receptor stimulated by S1P, can activate PI3K [48] (Figure 1). While in bovine aortic ECs (BAECs), the S1P_1_ receptor activation was reported to activate PI3K–Akt in a different way [49]. Therefore, Gβγ downstream of the S1P_1_ receptor, successively activated the Src tyrosine kinase, Tiam1 [(T-lymphoma invasion and metastasis gene 1), a guanine nucleotide exchange factor (GEF) for Rac1], Rac1, PI3K, Akt, and endothelial nitric oxide synthase (eNOS) [49]. In HUVECs, adenosine receptor type 1 (A_1_AR), which is coupled to G_i_ proteins [28], enhances HUVECs’ survival via activation of the PI3K–Akt pathway [29] (Table 1). Thus, Gβγ, which can be activated due to ROS-induced activation of Gα_i_ (Nishida) [24,25] is likely to activate the PI3K–Akt pathway in a receptor-independent manner, and promote EC survival (Figure 1).

### 2.4. G_i/o_ Proteins and Pro-Apoptotic Pathways in ECs

It should be noted here that activation of the G_i/o_-coupled receptors may also lead to EC apoptosis (Table 1). In HUVECs, stimulation of cannabinoid receptors (CB1 and CB2 are both coupled to G_i/o_ proteins [30]) by anandamide, induced apoptosis via activation of the JNK and p38 MAPK [31]. In human coronary artery ECs (HCAECs), stimulation of the CB1 receptor led to apoptosis via increase in ROS generation and activation of JNK and p38 MAPKs [32].

## 3. ROS Can Mimic Insulin Signaling

Non-insulin-dependent diabetes mellitus (NIDDM) is regarded as a significant contributor to the development of endothelial dysfunction [50]. There are paradoxical interrelationships between ROS and insulin signaling. ROS are known to participate in insulin signal transduction, as well as to evoke insulin resistance [51,52,53,54].

### 3.1. Reversible Inhibition of Protein Tyrosine Phosphatase 1B (PTP1B) by ROS

Insulin induces activation of NADPH oxidases [53,54], and the generated ROS transiently inhibit protein tyrosine phosphatase 1B (PTP1B) [55]. Increased insulin sensitivity was observed in mice deficient of PTP1B [56,57]. In a mouse model of pancreatic islet transplantation into eye, deletion of PTP1B, elevated revascularization of the graft islet, increased graft survival, facilitated recovery of normoglycemia, and improved glucose tolerance [58]. These effects of loss of PTP1B were mediated via an increase in the expression of VEGF-A by β cells, following activation of the peroxisome proliferator-activated receptor γ coactivator 1α (PGC1α) and the estrogen-related receptor α [58].

### 3.2. Indirect Activation of the PI3K–Akt Pathway

There are several points where ROS can enhance the signal transduction through the PI3K–Akt module (Figure 2). Ras GTPases can be directly activated by ROS via oxidation of Cys118 [59]. In BAECs, peroxynitrite activated p21Ras, which in turn activated the PI3K–PDK1–Akt pathway [60]. Furthermore, CaMKII can directly phosphorylate and activate Akt [61]. CaMKII itself can be activated through a reversible oxidation of methionines 281/282 [62].

Insulin receptor substrate downstream 1 (IRS-1), an adaptor protein providing spatial organization of signaling proteins of the insulin receptor (IR), binds to the regulatory p85 subunit of PI3K, so that the catalytic p110 subunit of PI3K can convert phosphatidylinositol-4,5-bisphosphate (PIP2) into phosphatidylinositol-3,4,5-trisphosphate (PIP3) [63]. PIP3 serves the binding and activation of 3-phosphoinositide-dependent kinase 1 (PDK-1), which phosphorylates and activates its target Akt [63]. Phosphatase and the tensin homolog deleted on chromosome 10 (PTEN), dephosphorylates PIP3, thus, interrupting signal transduction onto Akt [64]. Exposure to H_2_O_2_ can result in the formation of a disulfide bridge between Cys124 and Cys71 in PTEN [65], leading to its inactivation, which results in an indirect activation of the PI3K–Akt pathway by H_2_O_2_ [65,66,67,68].

In addition, active Akt can be deactivated through dephosphorylation of Ser473 and Thr308 by protein phosphatase 2A (PP2A) [61,69,70]. PP2A can be reversibly inhibited by ROS through disulfide bond formation in the catalytic subunit of PP2A [69], which would enhance the signal transduction through the PI3K–Akt pathway (Figure 2).

### 3.3. Small GTPase Ras and Ras–Raf–MEK–ERK Pathway

The small GTPase Ras is coupled to receptor tyrosine kinases (RTKs) via the adaptor protein Grb2 (growth receptor binding protein 2) and GEF Sos (son-of-sevenless) [71,72]. Ras activates serine/threonine kinase Raf, which in turn activates MEK (MAP/ERK kinase), and MEK downstream activates ERK [73]. In 3T3-L1 adipocytes, Nox4 generated H_2_O_2_-mediated insulin-induced activation of Erk [54]. Ras–Raf–MEK1/2–ERK1/2 is a signaling branch downstream of RTKs, including the insulin receptor and the VEGF receptor, for example [71,72,74]. There are four isoforms of the small GTPase Ras—H-Ras, N-Ras, K-Ras4A, and K-Ras4B [59]. Oxidation of Cys118 located in the NKCD (Asn116-Lys117-Cys118-Asp119) amino acid sequence is responsible for activation of Ras by ROS [59]. In BAECs, peroxynitrite activated p21Ras via *S*-glutathiolation of Cys118, which led to the activation of the Raf-1–MEK–ERK and the PI3K–PDK1–Akt pathways [60]. In addition, Cys80, Cys181, Cys184, and Cys186 may also participate in redox regulation of Ras [59]. However, in BAECs, oxidation of Cys181/184 in H-Ras impaired its palmitoylation and plasma membrane localization, and led to apoptosis [74].

## 4. Endothelial Barrier: Redox Dependence of Some Intracellular Signaling Proteins Involved in Regulation of Endothelial Permeability

Generally, oxidative stress leads to an increase in endothelial permeability [75]. Here, we discuss data suggesting that ROS can also contribute to the stabilization of the endothelial barrier. Among multiple pathways implicated in the regulation of the endothelial barrier [76], some can be regulated by ROS.

### 4.1. ROS and G_i/o_ Proteins

Hydroxyl radicals are likely to activate Gα_i/o_ in a G-protein-coupled receptor (GPCR)-independent manner [25]. It appears that GPCR-dependent activation of Gα_i/o_ proteins has dual effects on endothelial permeability. For example, the interleukin 8 chemokine receptor CXCR2 is coupled to the G_i/o_ proteins [77] and can increase pulmonary microvascular permeability in a murine model of LPS-induced lung injury [78]. Similarly, human cerebral microvascular ECs impaired their barrier integrity, upon activation of CXCR2 [79].

On the other hand, there is experimental evidence for the role of Gα_i_ in endothelial barrier stabilization. Active Gβγ, downstream of the S1P_1_ receptor, activated the PI3K–Akt pathway [48,49], which can enhance the endothelial barrier [80] (Figure 3). In calf pulmonary artery vasa vasorum ECs (VVECs), activation of the G_i/o_-coupled A_1_AR, increased the endothelial barrier integrity via activation of the PI3K–Akt pathway [81]. There can be at least three pathways that lead from active Gβγ subunits to activation of Rac1, a small GTPase known to stabilize the endothelial barrier [76,82,83,84] (Figure 3). In BAECs, active Gβγ dimers downstream of S1P_1_, can activate the Src–Tiam1–Rac1–PI3K–Akt pathway [49] (Figure 3). In HPAECs, Gβγ subunits downstream of the S1P_1_ receptor, activated the PI3K–Akt–Src–Tiam1–Rac1 pathway [48] (Figure 3). In calf pulmonary artery VVECs, G_i_ proteins downstream of A_1_AR, appear to activate the SHP2 (Src homology region 2 domain-containing phosphatase-2)–Rac1–PKA pathway and to induce remodeling of the actin cytoskeleton [82] (Figure 3).

Moreover, there seems to be a cAMP-independent pathway leading from the activation of the G_i_ proteins to the activation of PKA and endothelial barrier enhancement [85,86] (Figure 3). In HPAECs, activation of P2YRs by ATP and its analogue adenosine 5′-[γ-thio]-triphosphate (ATPγS), coupled to the G_q_ and G_i_ proteins, and led to an enhancement of the endothelial barrier [85]. It appears that G_i_ proteins can activate PKA in an AC-independent manner via the PKA-anchoring proteins (AKAPs), and PKA phosphorylates VASP (vasodilator-stimulated phosphoprotein) and thus enhances the barrier [85]. Similarly, in human lung microvascular ECs, adenosine and ATPγS stabilized the endothelial barrier via P2Y_4_R (coupled to both G_q_ and G_i_) and P2Y_12_R (coupled to G_i_), and an unconventional cAMP-independent activation of PKA [86].

### 4.2. ROS and Some Branches of Signaling Downstream of Growth Factor Receptors

In bovine pulmonary artery endothelial cells (BPAECs), activation of the MEK–ERK pathway led to an increase in the endothelial permeability [41,87]. ROS may contribute to the activation of the Ras–Raf–MEK–ERK pathway via a direct activation of Ras [59,74]. In rat aortic vascular smooth muscle cells, ERK1/2 was activated by CaMKII [88], which itself could be activated by ROS [62].

In rat coronary microvascular endothelial cells, insulin was shown to stabilize the endothelial barrier via the PI3K–Akt pathway [80]. ROS can enhance PI3K–Akt signaling via activation of Ras and CaMKII [59,62,74] and inactivation of PTEN and PP2A [65,66,69].

The small GTPase Rac1, which stabilizes the endothelial barrier [76,80,84], can be directly activated by ROS [89,90].

The small GTPase RhoA can be directly inhibited by oxidants [89,90]. RhoA via activation of its downstream effector Rho-associated kinase (ROCK), induces phosphorylation of the myosin light chain kinase (MLCK), which activates non-muscle myosin II [91], resulting in EC contraction and increased endothelial permeability [76,83]. Inhibition of myosin phosphatase by ROCK also contributes to non-muscle myosin II activation [92].

### 4.3. ROS and Ca^2+^-Dependent Mechanisms

Stimulation of Ca^2+^-dependent pathways can increase endothelial paracellular permeability [76]. For example, activation of CaMKII in bovine pulmonary artery ECs disrupted the endothelial barrier via phosphorylation of caldesmon and activation of the ERK [87].

Increase in intracellular Ca^2+^ may occur via opening of the Ca^2+^ channels of the plasma membrane, release of Ca^2+^ from intracellular stores through ryanodine receptors and inositol-trisphosphate (IP3) receptors, or from mitochondria ― all these mechanisms being regulated by ROS [93]. ROS can also activate protein kinases such as PKA type I [94], protein kinase C [95], and CaMKII [62]. These kinases regulate both the ion channels [93] and the endothelial permeability [76,84].

### 4.4. ROS and Tyrosine Kinases and Phosphatases

The effect of Src on the endothelial barrier appears to be biphasic—initially Src can enhance the barrier function, but prolonged action of Src impairs the endothelial barrier [96]. H_2_O_2_ can activate Src tyrosine kinase via reversible sulfenylation (Cys-SOH formation) of two cysteine residues—Cys-185 and Cys-277 [97].

In HPAECs, c-Abl enhances the endothelial barrier, apparently via regulation of the actin cytoskeleton [98]. Exposure of COS7 cells to high concentrations (1 mM) of H_2_O_2_, induced a 5-fold increase in c-Abl tyrosine kinase activity [99].

In several model systems, including a mouse model of acute LPS-induced lung injury, HUVECs, and HPAECs, Lyn tyrosine kinase was revealed to stabilize the endothelial barrier [100]. Lyn can be directly activated by H_2_O_2_ [101].

Inhibition of PTP1B can increase paracellular endothelial permeability via an increase in phosphorylation of vascular endothelial (VE)-cadherin and disruption of cell–cell adhesions [102]. PTP1B can be inhibited by ROS [55].

## 5. Endothelial Dysfunction

Endothelial dysfunction is characterized by 3 main features—impaired flow-induced endothelium-dependent vasodilatation, increased pro-inflammatory activity of endothelium, and enhanced pro-thrombotic status of endothelium [103]. Oxidative stress appears to be a key factor in the pathogenesis of endothelial dysfunction [103]. ECs are exposed to fluid shear stress (SS), which, depending on its pattern, can be either laminar shear stress (LSS), well-established as the anti-inflammatory and athero-protective factor, or disturbed shear stress (DSS), which is known to lead to vascular inflammation and atherosclerosis [104,105,106,107]. H_2_O_2_ produced by ECs is itself regarded as an endothelium-derived hyperpolarizing factor, since it directly activates the PKGIα in vascular smooth muscle cells [108,109]. It is of interest to see if and how ROS could ameliorate endothelial dysfunction. For this purpose, we first discuss the potential involvement of ROS in the production of NO by ECs. Next, we regard the role of the mechano- as well as the redox-sensitive MEK5–ERK5 pathway, in the regulation of transcription factors that are known to suppress inflammation.

Since endothelial dysfunction comprises impairment of several EC functions, agents that possess a wide spectrum of actions are of interest as therapeutics. Phytochemicals can serve as agents that target multiple signaling mechanisms in ECs [110,111]. In vivo studies have shown such effects of indole-3-carbinol (I3C), a phytochemical found in cruciferous vegetables, and its derivative 3,3,’-diindolylmethane (DIM) on EC functions such as suppression of angiogenesis, prevention of thrombus formation, and alleviation of inflammation. Suppression of the inflammatory response by I3C and DIM are mediated via suppression of production and release of the inflammatory cytokines, modulation of ROS production, inhibition of leucocyte–EC interaction, and decrease in paracellular microvascular permeability. It appears that I3C and DIM can exert dual effects on ROS production; both stimulation and inhibition of ROS generation were reported [110]. Maslinic acid, a triterpene derivative from *Olea europaea*, was shown to suppress activation of NF-κB in human dermal microvascular ECs and human placenta-derived pericytes. Suppression of NF-κB by maslinic acid reduced the expression of adhesion molecules E-selectin, intercellular adhesion molecule 1, and vascular adhesion molecule 1 on EC and pericytes, and attenuated the development of inflammation [111].

### 5.1. ROS and Flow-Induced Release of Vasodilators by ECs

Here, we regard three scenarios in which, hypothetically, ROS can promote activation of eNOS and release of NO—via redox sensitive CaMKII-, PI3K–Akt-, and G_i/o_-mediated pathways.

Fluid SS activates numerous K^+^, Na^+^, Ca^2+^, and non-selective ion channels in ECs [112] (Figure 4). A number of K^+^ and Ca^2+^ ion channels are themselves directly regulated by ROS [113]. For example, the TRPC6 (transient receptor potential canonical 6) channel, which is known to be activated by mechanical stress [114,115], can also be activated by H_2_O_2_ [116,117]. H_2_O_2_ can activate the L-type Ca^2+^ channels [118]. In addition, ion channels are regulated by redox-sensitive protein kinases like PKA, PKC, and CaMKII [62,93,94,95]. Increase in [Ca^2+^]_i_ activates CaMKII, but ROS can also activate CaMKII via oxidation of methionines 281/282 in a Ca^2+^-independent manner [62]. Active CaMKII phosphorylates and activates Akt [61]. Active Akt can phosphorylate and activate eNOS [46]. PP2A can inhibit Akt by dephosphorylation [61,69,70]. Additionally, PP2A inhibits eNOS via its dephosphorylation at Ser1177 [119]. Inhibition of PP2A by ROS [69] can, therefore, facilitate the activation of Akt and eNOS (Figure 4). Thus, in case of stimulation of NO production by SS via sequential activation of Ca^2+^ channels, CaMKII, Akt, and eNOS, ROS can contribute to the activation of all four entities—Ca^2+^ channels, CaMKII, Akt, and eNOS (Figure 4).

In ECs, SS stimulates Akt, which activates endothelial NO synthase (eNOS) by phosphorylation at Ser1177 [120] or Ser 1179 [121] in a Ca^2+^-independent manner [46]. In this mechanism, ROS can contribute to Akt activation (see Section 3.2).

There can be several pathways from G_i_ proteins to the activation of eNOS. For example, stimulation of G_i_-coupled S1P_1_ receptor appears to activate the following signaling cascade: Gβγ–Src–Tiam1–Rac1–PI3K–Akt–eNOS [49] (Figure 5). G_i_ proteins can be activated by both SS [122,123] and ROS [24,25], suggesting that ROS can enhance or even mimic SS-induced G_i_ activation in ECs (Figure 5).

Furthermore, some steroid hormone receptors are well-established to be coupled to G_i/o_ proteins and activate eNOS [124,125]. Endogenous estrogens act via three different receptors—classical estrogen receptors α and β (ERα and ERβ) and G-protein-coupled estrogen receptor (GPER), also known as GPR30 [126]. In immortalized ovine pulmonary artery endothelial cells (iPAECs), the plasma membrane ERα was shown to localize to caveolae and to stimulate eNOS [124,127] via activation of Gα_i_ [124]. In BAECs, membrane receptor of the adrenal dehydroepiandrosterone (DHEA) was shown to activate eNOS via coupling to Gα_i2_ and Gα_i3_, but not Gα_i1_ or Gα_o_ [125]. In this scenario, ROS can contribute to eNOS stimulation via direct activation of the G_i/o_ proteins [24,25].

### 5.2. ROS, MEK5–ERK5 Module and Transcription Factors (TFs) That Can Alleviate Endothelial Dysfunction

Inflammation is regarded as a key factor in the pathogenesis of atherosclerosis [128,129,130]. ROS are well-known to promote inflammatory response of ECs [131]. LSS-dependent EC survival and quiescence are mediated by TFs, such as KLF2 (Krüppel-like factor 2), KLF4, and Nrf2 [106]. The DSS-induced pro-inflammatory response of ECs on the other hand is evoked by TFs, such as NF-κB, AP-1, YAP/TAZ, and HIF-1α [106]. Let us regard several hypothetical scenarios where ROS may contribute to mechanisms that prevent endothelial dysfunction via regulation of athero-protective TFs (Figure 6).

#### 5.2.1. Redox-Sensitive Elements in SS-Induced Signaling Pathway Leading to ERK5 Activation

To illustrate how ROS can influence the signal transduction on athero-protective TFs, we have chosen the ERK5 [also known as big mitogen-activated protein kinase 1 (BMK1)] and its upstream kinase MEK5 (MAPK/ERK kinase 5) as a signaling module that integrates both SS-induced athero-protective signaling [132,133] and redox sensitivity [134,135]. In BAECs, SS (12 dynes/cm^2^) activated ERK5 in a Ca^2+^-dependent and -Src-tyrosine-kinase-independent manner [136]. Mechano-sensitivity of the MEK5–ERK5 module was also demonstrated in HUVECs where LSS (12 dynes/cm^2^) activated ERK5 [133,137]. Since Ca^2+^-dependent mechanisms are involved in the activation of ERK5 in BAECs [136], ROS can interfere in these signaling proteins via effects on the Ca^2+^-permeable channels [93] and via direct activation of CaMKII [62].

#### 5.2.2. Krüppel-Like Factors (KLF) Family

TFs of the Krüppel-like factor (KLF) family include 17 members, among which KLF2, KLF4, and KLF6 are expressed in ECs [138,139]. KLF2 and KLF4 exert anti-inflammatory, athero-protective, and anti-thrombotic functions in ECs [138,139]. Furthermore, in HUVECs, KLF2 contributes to regulation of vascular tone via downregulation of endothelin-1 and adrenomedullin, and upregulation of eNOS [140]. In various types of ECs, athero-protective laminar flow activates the MEK5–ERK5 pathway, which activates TF MEF2 (myocyte enhancer family 2), and MEF2 induces the transcription of KLF2 [132]. In HUVECs, laminar SS activated KLF2 via the MEK5α–ERK5 pathway [133].

#### 5.2.3. MEF2 Family of Transcription Factors

The MEF2 family of transcription factors includes four members—MEF2A, MEF2B, MEF2C, and MEF2D [141]. Among these factors MEF2A, MEF2C, and MEF2D, but not MEF2B, can be phosphorylated and activated by BMK1 [141]. In human retinal ECs and HUVECs, MEF2C was shown to inhibit tumor necrosis factor alpha (TNF-α)-induced activation of NF-κB, to suppress the expression of pro-inflammatory genes and to decrease leukocytes adhesion to ECs [142]. In PC12 cells, H_2_O_2_ activated the c-Src–MEK5–ERK5–MEF2C pathway [134], suggesting that in ECs, H_2_O_2_ is also likely to activate MEF2C and contribute to the inhibition of inflammation.

#### 5.2.4. Nuclear Factor Erythroid 2-Related Factor 2 (Nrf2)

Nrf2 is a key transcription factor, which is activated by oxidants and turns on the cellular defense against oxidative stress [143,144,145]. In addition, Nrf2 can also alleviate inflammation [146].

At physiological ROS levels, Nrf2 is kept inactive and directed to proteasomal degradation via association with the redox sensor KEAP1 (Kelch-like Ech-associated protein 1) [147]. ROS induce dissociation of the Nrf2-KEAP1 complex and Nrf2 is then translocated into the nucleus [147]. Nrf2 is a key TF mediating cyto-protection against oxidative stress in HUVECs [133]. In HUVECs, laminar SS (12 dynes/cm2) was shown to activate Nrf2 via activation of MEK5α and its effector ERK5 [133].

Normal level of ROS is well-known to provide antioxidant defense in ECs via stimulation of Nrf2, which activates antioxidant response element (ARE)-dependent expression of antioxidant genes. Nrf2-dependent vascular protection also includes the suppression of the NF-κB-dependent pro-inflammatory pathway and improvement of the mitochondrial function [148]. Recently, in an in vitro model of atherosclerosis—exposure of human coronary artery endothelial cells (HCAECs) to oxidized low-density lipoprotein (ox-LDL)—activation of Nrf2 by prenyldiphosphate synthase subunit 2 (PDSS2) was reported. Activation of Nrf2 decreased ferroptosis and promoted proliferation of HCAECs [149]. In HUVECs, activation of Nrf2 by 1,25 dihydroxyvitamin D3 was shown to protect against high glucose-induced injury [150].

#### 5.2.5. Peroxisome Proliferator-Activated Receptors (PPARs)

The family of nuclear receptors PPARs are presented by three subtypes—PPARα, PPARβ/δ, and PPARγ [151]. PPARγ plays important roles in the regulation of lipid metabolism, insulin resistance, vascular inflammation, and arterial hypertension [151]. Expression of a constitutively active mutant of PPARγ1 in HUVECs, suppressed the activation of pro-inflammatory TFs AP-1 and NF-κB, resulting in a reduced expression of markers of inflammation, such as ICAM-1, VCAM-1, and E-selectin [152]. In HUVECs, ERK5 mediated flow-induced activation of PPARγ1 [137], suggesting that H_2_O_2_ ― via activation of the c-Src–MEK5–Erk5 pathway [134,135]―can also activate PPARγ1. Furthermore, ROS may induce activation of PPARγ via generation of oxidized fatty acids, which were shown to bind to and activate PPARγ [153].

## 6. Role of NADPH Oxidase-Derived ROS in Angiogenesis

There is increasing evidence that ROS derived from NADPH oxidases significantly contribute to signaling mechanisms regulating angiogenesis, a process of formation of new blood vessel from the pre-existing vessel. ECs’ proliferation, migration, differentiation, and capillary tube formation constitute the main features of angiogenesis. Angiogenesis is important for embryonic development, wound healing, and post-ischemic neovascularization. Endothelial NADPH oxidases, particularly NOX2 and NOX4, produce ROS that play important role in the regulation of angiogenesis [7,121,154,155,156,157,158,159,160,161,162]. Endothelial ROS generation itself is subject to tight spatial and temporal regulation [156,163,164]. Endothelial NADPH oxidases can be activated by growth factors, cytokines, ligands of GPCRs, mechanical forces, and metabolic factors. Downstream, NADPH-derived ROS participate in several cellular processes, such as regulation of self-renewal, survival, proliferation, and differentiation of mesenchymal stem cells. Moreover, commitment of stem cells to adipogenic, osteogenic, or myogenic lineage, and endothelial–mesenchymal stem cell transition are also ROS-dependent [7]. Upstream inducers of pro-angiogenic NOX4 include hypoxia, ischemia, VEGF, TNF-related apoptosis-inducing ligand (TRAIL), and transforming growth factor-β1 (TGF-β1) [160].

ROS derived from NOX1 and NOX4 are important regulators of proliferation, hypertrophy and apoptosis in human pulmonary artery endothelial and smooth muscle cells, which can lead to airway and vascular remodeling. Lung airway and vascular remodeling lead to disorders such as pulmonary artery hypertension, chronic obstructive pulmonary disease, asthma, and neonatal bronchopulmonary dysplasia. Factors inducing NADPH oxidases and increased ROS generation in lung include hyperoxia, hypoxia, LPS, allergens, angiopoietin-2, EGF, TGF-β, bone morphogenetic proteins, interleukins, and S1P. Downstream, ROS activate transcription factors such as NF-κB and AP-1, which results in the development of inflammation and vascular cell proliferation [158]. In case of peripheral artery disease, physiological levels of NADPH oxidase-derived ROS are pro-angiogenic and can stimulate EC proliferation, sprouting, migration, and tubule formation. Additionally, ROS contribute to the stability of newly-formed vessels via regulation of pericytes [159]. Pathogenesis of pulmonary arterial hypertension involves endothelial dysfunction accompanied by smooth muscle cell proliferation, inflammation, and fibrosis. The main underlying cause may be excessive production of ROS by NADPH oxidases (NOX1, NOX2, NOX4) and mitochondria [121].

Surprisingly, both ROS generation by NOX4 and ROS scavenging by thioredoxin 2 (TRX2) can promote angiogenesis. TRX2, a key mitochondrial ROS scavenger, promote EC survival and proliferation via elimination of ROS and thus enhancement of NO availability, and also via inhibition of apoptosis signaling kinase-1 (ASK1). The mechanism of cross-talk between NOX4 and TRX2 may depend on regulation by common angiogenic factors, such as hypoxia, ischemia, and VEGF [160], or spatial presence of NOX4-derived ROS and absence of mitochondrial ROS.

Effects of NADPH-derived ROS on angiogenesis depend on isoform of NADPH oxidase. NOX4-generated ROS can promote vascular restoration after hypoxic and ischemic injuries and can inhibit vascular inflammation. However, in case of oxygen-induced retinopathy, NOX4 promotes pathological angiogenesis. NOX2 appears to be involved not only in normal physiological angiogenesis but also in pathological angiogenesis, in cases of choroidal neovascularization, retinopathy, and tumor growth. Moreover, NOX2-generated ROS were shown to inhibit physiological angiogenesis via activation of apoptotic signaling in the retina and the brain, but promotes vascular restoration in ischemic hindlimb. Additionally, activation of NOX2 by proinflammatory TNFα can contribute to vascular inflammation [161]. NOX2-generated ROS were shown to significantly contribute to diabetes-induced premature senescence of retinal ECs [165].

NADPH-produced ROS exert their actions on angiogenesis via regulation of intracellular signaling and gene expression. ROS regulate signaling proteins via reversible oxidation of cysteine residues to sulfenic acid (-SOH), sulfinic acid (-SO_2_H), and sulfonic acid (-SO_3_H). Protein tyrosine phosphatases contain a conserved cysteine residue in their catalytic domain and oxidation of this cysteine to sulfenic acid and sulfonic acid reversibly and irreversibly, respectively, inhibit tyrosine phosphatases [162].

VEGF, a key governor of angiogenesis, regulates angiogenesis mainly via VEGFR2 [166,167]. VEGF via VEGFR2 and Rac1 stimulates ROS production by Nox2 in ECs [163]. VEGF via VEGFR2 and Rac1 stimulates ROS production in ECs [163,168]. Signal transduction through the VEGFR2 is facilitated by reversible ROS-induced inhibition of protein phosphatases such as SHP1 [169], LMW-PTP [170], PTP1B, and density-enhanced phosphatase-1 (DEP1) [171]. VEGFR2 induces Nox2-dependant ROS production, leading to a localized formation of cysteine sulfenic acid in IQCAP1 protein (Cys-OH-IQCAP1), at the leading edge of migrating EC [167], thereby promoting directional EC migration.

Finally, ROS trigger mobilization of bone marrow progenitor cells in response to ischemic injury. In a mouse model of hindlimb ischemia, Nox2-derived ROS were shown to regulate the mobilization of progenitor cells from the bone marrow. Effects of hindlimb ischemia-induced Nox2-derived ROS included increase in expression of HIF-1α and VEGF throughout the bone marrow, elevated survival and proliferation of bone marrow Lin- progenitor cells, Akt phosphorylation, activation of matrix metalloproteinase-9 (MMP-9), and membrane type 1-MMP (MT1-MMP) [172].

Since mitochondria also generate ROS, the cross-talk between NADPH oxidases, and mitochondria, the process referred to as ROS-induced ROS release (RIRR), contribute to VEGF- and angiopoietin-1-induced angiogenesis [173,174,175].

## 7. Conclusions

The aim of this review was to analyze the literature evidence to test our hypothesis that ROS can evoke beneficial effects on ECs. Analysis of the available experimental data has shown that ROS can mimic, to some extent, signaling through the G_i_-protein coupled receptors, as well as insulin receptor and other growth factor receptors. Although, the prevalent view is that ROS are responsible for an increase in paracellular endothelial permeability, accumulating data suggest that physiological ROS may be actively contributing to endothelial barrier stabilization or maintenance. Development of endothelial dysfunction involves numerous signaling proteins, and a few of them can be either activated or inhibited by ROS. In particular, vascular inflammation is one of primary conditions leading to vascular diseases. Of great interest would be to explore the causal relationships between ROS and the activities of transcription factors that regulate inflammatory response. Further investigations are needed to better delineate the boundary between normal physiological and pathological ROS levels. As ROS are tiny and short-lived, their live-cell, real-time observation is difficult. Nevertheless, the progress in the field depends on studies of complexes between sources of ROS and targets of ROS.

## Figures and Tables

**Figure 1 antioxidants-10-00904-f001:**
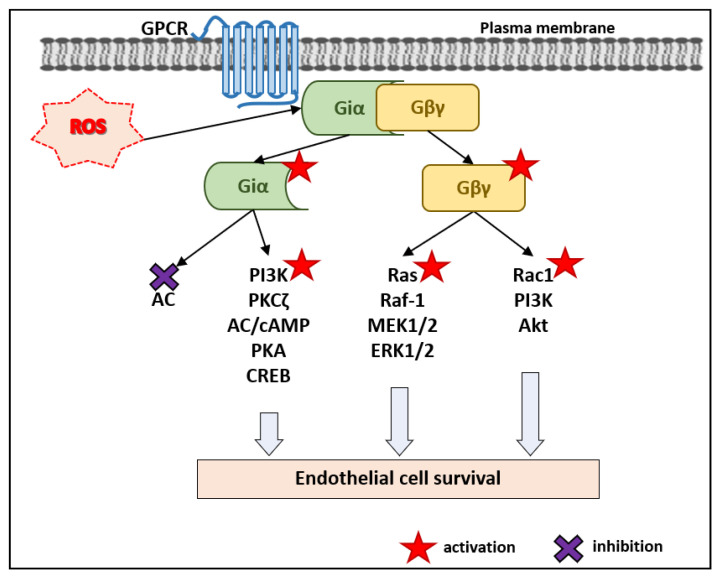
Scheme illustrating the potential mechanisms for activation of the pro-survival pathways by ROS-induced activation of G_i_ proteins—both Gα_i_ and Gβγ subunits—in a G_i_-PCR-independent manner.

**Figure 2 antioxidants-10-00904-f002:**
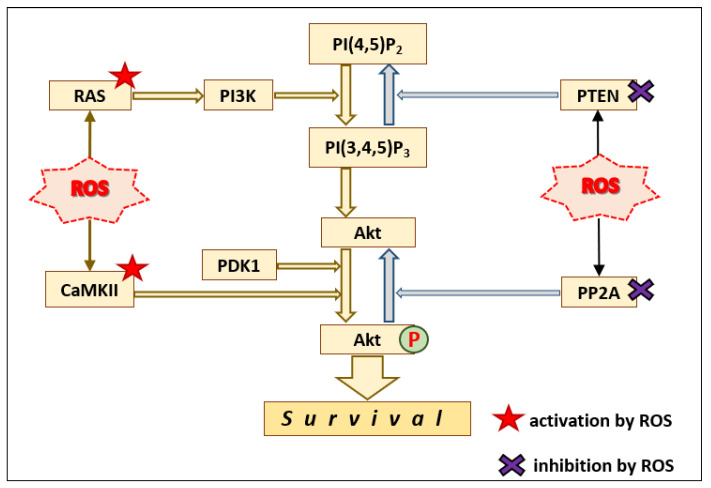
Potential mechanisms for indirect activation of the PI3K–Akt pathways by ROS-induced activation of Ras and CaMKII, as well as ROS-induced inhibition of PP2A.

**Figure 3 antioxidants-10-00904-f003:**
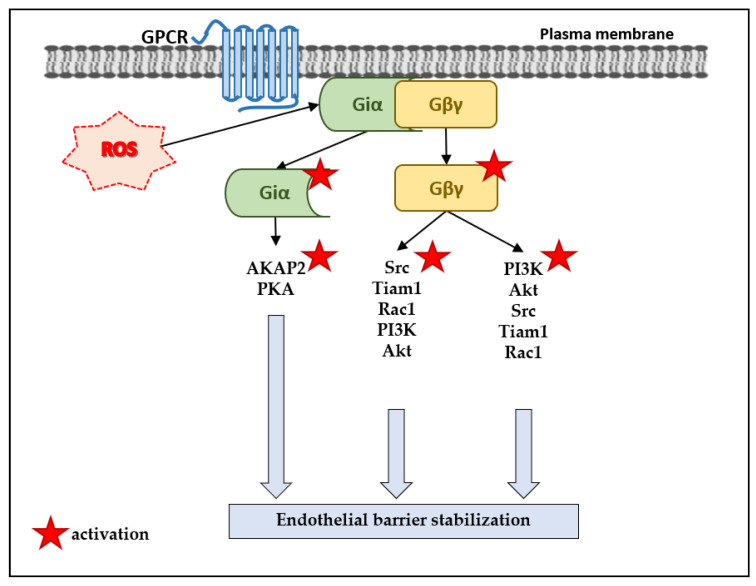
Scheme illustrating the potential mechanisms for enhancement of the endothelial barrier via ROS-induced and G_i_-PCR-independent activation of the Gα_i_ proteins and liberation activation of the Gβγ dimers, which can further activate the AKAP2–PKA, Rac1, and the PI3K–Akt pathways.

**Figure 4 antioxidants-10-00904-f004:**
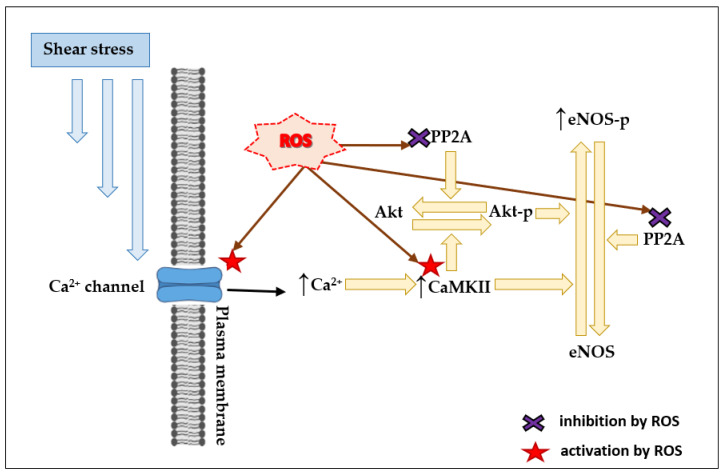
Scheme illustrating the potential involvement of ROS into flow-induced CaMKII-mediated eNOS activation.

**Figure 5 antioxidants-10-00904-f005:**
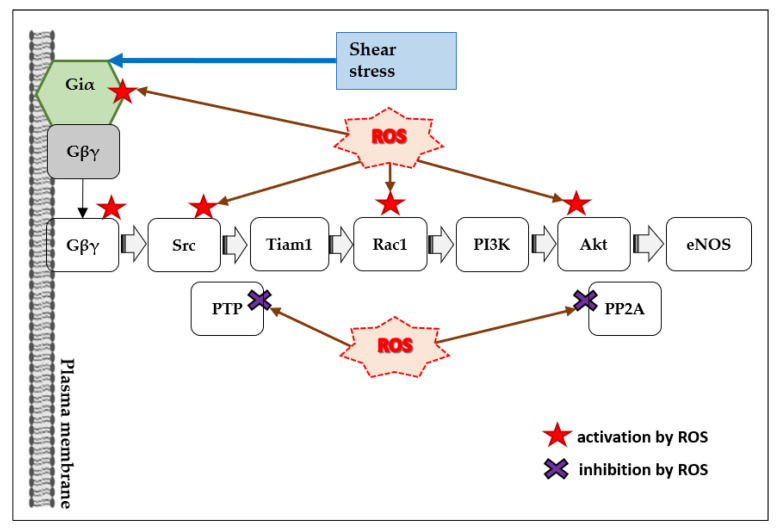
Scheme illustrating the potential involvement of ROS into flow-induced G_i_-mediated eNOS activation.

**Figure 6 antioxidants-10-00904-f006:**
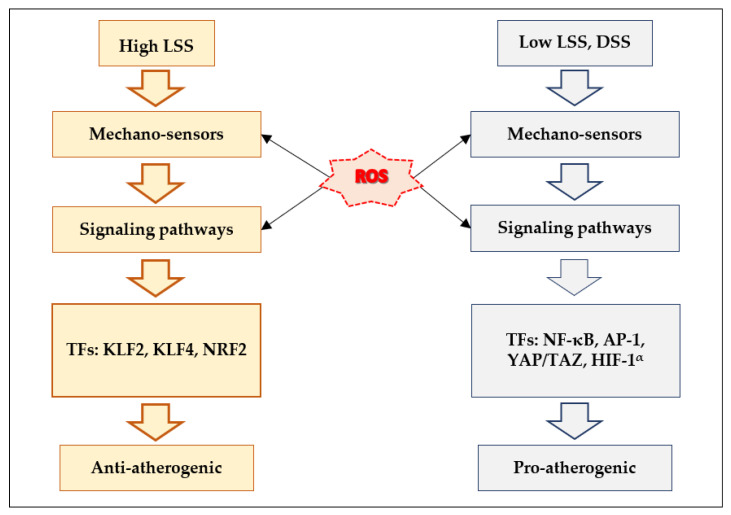
Scheme illustrating the potential influence of ROS on shear-stress-induced activation of transcription factors involved in anti-atherogenic and pro-atherogenic responses. LSS: laminar shear stress; DSS: disturbed shear stress; and TFs: transcription factors.

**Table 1 antioxidants-10-00904-t001:** Involvement of Gi proteins, activated by ligand binding to GPCRs, in regulation of EC survival, and apoptosis.

**Agonist**	**Receptor(s)**	Coupling of Receptor to G_i_ Proteins	Effect on EC Survival and Apoptosis		
			Pro-Survival or Pro-Apoptotic, EC Type	Signaling Pathway	Reference(s)
Adenosine	A_1_AR	[28]	Pro-survival, HUVEC	PI3K—Akt	[29]
Anandamide	CB1, CB2	[30]	Pro-apoptotic, HUVEC	Activation of JNK and p38 but not ERK	[31]
Anandamide	CB1	[30]	Pro-apoptotic, HCAEC	JNK and p38	[32]
Ghrelin gene products	GHS-R1a	G_q/11_, G_i_ [33]	Pro-survival, human pancreatic islet microvascular ECs	AC–cAMP–PKA	[27]
S1P	S1P_1_	G_i_ [34]	Pro-survival, HUVEC	ERK1/2	[35]

HUVECs, human umbilical vein endothelial cells; HCAEC, human coronary artery ECs.

## Data Availability

Data is contained within the article.

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
