# Peer review of "Pleiotropic and Potentially Beneficial Effects of Reactive Oxygen Species on the Intracellular Signaling Pathways in Endothelial Cells"

_antioxidants, 2021, doi:10.3390/antiox10060904_

Round 1
Reviewer 1 Report
In this paper entitled “Oxygen as a primary pro-survival signal for endothelial cells when converted into reactive oxygen species”, Dr. Pantaleo and co-workers review and analyze the available experimental data on ROS activation of pro-survival signaling pathways in ECs.
ROS are important signaling molecules known to induce oxidative stress and contribute to pathogenesis of many diseases. In this paper they analyze the available data to suggest, that ROS are important not only as toxic agents but also may play a permissive role for the majority of signaling pathways in ECs.
This review paper is very interesting, clearly written, the data are nicely presented and supporting their conclusions.
Author Response
We really appreciate the reviewer's positive feedback and we would like to thank the reviewer for the time spent on reviewing our manuscript.
Reviewer 2 Report
The article presents a synthesis of how ROS signals participate in some signaling pathways of endothelial cells. The article broadly describes how ROS signals participate in different fundamental pathways for the function of endothelial cells. Along these lines, I suggest that you modify the article's title for one more general to the function of endothelial cells since it is not only focused on the survival of endothelial cells as the current title suggests. Also, consider making a more elaborate conclusion, I believe that the review has great potential, and the conclusion is flawed compared with all the information previously provided.
Author Response
We would like to thank the reviewer for the time spent on reviewing our manuscript and for his suggestions helping us improving the article.
We have changed the title to: “Pleiotropic and potentially beneficial effects of reactive oxygen species on the intracellular signaling pathways in endothelial cells”, We have also modified the conclusion section accordingly.
Reviewer 3 Report
The review of Barvitenko and colleagues is a detailed description of how ROS can influence endothelial functions. The authors have made a huge effort to summarize the known mechanisms of the different cellular levels, however the review sound like a stringing together in some section. Therefore, I have some suggestion, which should be addressed by the authors to improve their review.
1.) Title is misleading because the readers expect detailed analyses of pro- and anti-apoptotic pathways depending on endothelial ROS levels. However, this review is more focused on signaling influencing pro- and anti-apoptotic pathways. Hence, the title should be rephrased.
2.) The sentence “Critical roles of ROS in angiogenesis are discussed in reviews in the Special Issue of Antioxidants “ROS Derived from NADPH Oxidase (NOX) in Angiogenesis” [7, 23–29].” should be removed as it gives the impression that these reviews are more suitable for a general overview.
3.) In line 155 “(CB1 and CB2 are both coupled to Gi/o proteins [37])” the brackets are set incorrectly
4.) In line 165 (section 3.1): There is a well performed study of Figueiredo et al. (“Targeting pancreatic islet PTP1B improves islet graft revascularization and transplant outcomes”). In this study, they analyzed the role of the phosphatase in islet endothelial cells. This information should be include.
5.) The information from lines 208-2017 has nothing to do with endothelial cells. Therfore, it should be removed or at least the authors should contextualize.
6.) Throughout the whole review there are many short sections describing only one or two findings. For instance, section 4.2-4.4. The authors should combine all these sections to improve the reading flow.
7.) In line 405-413: Nrf2 is a major transcription factor in ROS signaling, and many study reported the effect of ROS on Nrf2-signaling in endothelial cells. For instance PMID: 33946068, 33929387, 20524845…. Why the authors did not discussed these findings in this section?
8.) The section 5 (effect of ROS on endothelial dysfunction), the authos only discussed in vitro studies, however, the main features as suggested by the reviewer are the impaired flow-induced endothelium-dependent vasodilatation, increased pro-inflammatory activity of endothelium and enhanced pro-thrombotic status of endothelium. Accordingly, the authors have to include some animal studies showing that ROS affecte these parameters in vivo. For instance: PMID: 29532757 or 30992483.
Author Response
We express sincere gratitude to the Reviewer 3 for meticulous reading of our manuscript and helpful comments and suggestions.
1.) Title is misleading because the readers expect detailed analyses of pro- and anti-apoptotic pathways depending on endothelial ROS levels. However, this review is more focused on signaling influencing pro- and anti-apoptotic pathways. Hence, the title should be rephrased.
We have changed the title to: “Pleiotropic and potentially beneficial effects of reactive oxygen species on the intracellular signaling pathways in endothelial cells”
2.) The sentence “Critical roles of ROS in angiogenesis are discussed in reviews in the Special Issue of Antioxidants “ROS Derived from NADPH Oxidase (NOX) in Angiogenesis” [7, 23–29].” should be removed as it gives the impression that these reviews are more suitable for a general overview.
We have removed this sentence
3.) In line 155 “(CB1 and CB2 are both coupled to Gi/o proteins [37])” the brackets are set incorrectly
We have set the brackets correcty
4.) In line 165 (section 3.1): There is a well performed study of Figueiredo et al. (“Targeting pancreatic islet PTP1B improves islet graft revascularization and transplant outcomes”). In this study, they analyzed the role of the phosphatase in islet endothelial cells. This information should be include.
We have included information from Figueiredo et al., 2019 [ref. 58] into section 3.1.
5.) The information from lines 208-217 has nothing to do with endothelial cells. Therefore, it should be removed or at least the authors should contextualize.
We have removed these lines.
6.) Throughout the whole review there are many short sections describing only one or two findings. For instance, section 4.2-4.4. The authors should combine all these sections to improve the reading flow.
We thank the reviewer for pointing this out. We combined subsections 4.2-4.4. into subsection 4.2; we also combined subsubsections 5.1.1, 5.1.2 and 5.1.3 into one subsection 5.1. In some other cases, we believe that the points discussed in different (sub)subsections are so different (e.g., PI3K-Akt axis and Ca2+-dependent mechanisms) that is difficult to combine them under common subheading. Similarly, we believe that each discussed transcription factor deserves its own subsubsection.
7.) In line 405-413: Nrf2 is a major transcription factor in ROS signaling, and many studies reported the effect of ROS on Nrf2-signaling in endothelial cells. For instance, PMID: 33946068, 33929387, 20524845…. Why the authors did not discuss these findings in this section?
We have now added this information on effects of ROS on Nrf2 signaling into subsection 5.2.4.
8.) The section 5 (effect of ROS on endothelial dysfunction), the authors only discussed in vitro studies, however, the main features as suggested by the reviewer are the impaired flow-induced endothelium-dependent vasodilatation, increased pro-inflammatory activity of endothelium and enhanced pro-thrombotic status of endothelium. Accordingly, the authors have to include some animal studies showing that ROS affect these parameters in vivo. For instance: PMID: 29532757 or 30992483.
We have included this information in section 5 (lines 309-324) accordingly to reviewer’s suggestion
Round 2
Reviewer 3 Report
NA
Author Response
We much appreciate reviewer's suggestion and we have now included the section 6 for "Role of NADPH oxidase-derived ROS in angiogenesis ".
We have also updated literature review, additional references (refs.154-176) have been included accordingly.
Changes are labeled in red